# First Detection and Genetic Characterization of Swine Orthopneumovirus from Domestic Pig Farms in the Republic of Korea

**DOI:** 10.3390/v15122371

**Published:** 2023-11-30

**Authors:** Jonghyun Park, Hye-Ryung Kim, Eon-Bee Lee, Sang-Kwon Lee, Won-Il Kim, Young S. Lyoo, Choi-Kyu Park, Bok Kyung Ku, Hye-Young Jeoung, Kyoung-Ki Lee, Seung-Chun Park

**Affiliations:** 1Laboratory of Veterinary Pharmacokinetics and Pharmacodynamics, Institute for Animal Biomedical Science, College of Veterinary Medicine, Kyungpook National University, 80 Daehak-ro, Daegu 41566, Republic of Korea; parkjh@knu.ac.kr (J.P.); eonbee@gmail.com (E.-B.L.); 2DIVA Bio Incorporation, Daegu 41519, Republic of Korea; gpfuddl25@knu.ac.kr; 3Institute for Veterinary Biomedical Science, College of Veterinary Medicine, Kyungpook National University, Daegu 41566, Republic of Korea; sklee10@knu.ac.kr (S.-K.L.); parkck@knu.ac.kr (C.-K.P.); 4College of Veterinary Medicine, Jeonbuk National University, 79 Gobong-ro, Iksan 54596, Republic of Korea; kwi0621@jbnu.ac.kr; 5College of Veterinary Medicine, Konkuk University, 120 Neungdong-ro, Gwangjin-gu, Seoul 05029, Republic of Korea; lyoo@konkuk.ac.kr; 6Animal and Plant Quarantine Agency, Gimcheon 39660, Republic of Korea; kubk@korea.kr (B.K.K.); jhy98@korea.kr (H.-Y.J.)

**Keywords:** genome sequencing, phylogenetic analysis, swine orthopneumovirus, pneumovirus

## Abstract

Novel swine orthopneumovirus (SOV) infections have been identified in pigs in the USA and some European countries but not in Asian countries, including South Korea, to date. The current study reports the first SOV infections in four domestic pig farms located in four provinces across South Korea. The detection rate of SOV in oral fluid samples using qRT-PCR was 4.4% (14/389), indicating the presence of the virus in pigs at commercial farms in Korea. Two complete genome sequences and one glycoprotein (G) gene sequence were obtained from SOV-positive samples. The complete genome analysis of KSOV-2201 and KSOV-2202 strains showed 98.2 and 95.4% homologies with a previously reported SOV, and the phylogenetic tree exhibited a high correlation with a previously reported SOV strain from the US and a canine pneumovirus (CPnV) strain from China. Based on the genetic analysis of the viral G gene, the murine pneumonia virus (MPV)-like orthopneumoviruses (MLOVs) were divided into two genogroups (G1 and G2). Seventeen CPnVs and two feline pneumoviruses were grouped into G1, while the Korean SOV strains identified in this study were grouped into G2 along with one SOV and two CPnVs. These results will contribute to expanding our understanding of the geographical distribution and genetic characteristics of the novel SOV in the global pig population.

## 1. Introduction

The genus *Orthopneumovirus* of the family *Pneumoviridae* currently includes the *Orthopneumovirus bovis* (i.e., bovine respiratory syncytial virus (BRSV)), *Orthopneumovirus hominis* (i.e., human respiratory syncytial virus (HRSV)), and *Orthopenumovirus muris* (i.e., murine pneumonia virus (MPV)) species (ICTV, 2022). However, a serological study conducted in Northern Ireland in 1998 demonstrated the presence of another orthopneumovirus in pigs, and a more recent study conducted in the USA confirmed the viral genome of this using metagenomic sequencing [1,2]. Their findings showed that the viral genomic sequence was highly similar to that of MPV or canine orthopneumovirus (formerly known as canine pneumovirus (CPnV)), which was first identified in 2010 among dogs with respiratory diseases in the USA [3]. This novel porcine virus was named swine orthopneumovirus (SOV), and subsequent serological studies confirmed the occurrence of SOV infections in pigs in France and Spain [3,4]. However, there have been no reports of SOV infections in pigs in Asian countries, including South Korea, to date.

The current study presents the first case of SOV infection in domestic pig farms in South Korea, which is, incidentally, also the first country in Asia to report the detection of SOV. Genetic characterization of the Korean SOV strain was carried out using genomic sequencing and phylogenetic analyses, with the aim of providing greater insight into the molecular epidemiology of SOV infections in the global pig population.

## 2. Materials and Methods

### 2.1. Sample Collection and RNA Extraction

In 2022, 389 oral fluid samples were collected from 40 commercial farrow-to-finish pig farms suffering from respiratory distress in nine provinces in South Korea as part of the porcine reproductive and respiratory syndrome virus (PRRSV) surveillance project. Appendix A summarizes the characteristics of the pig farms examined in this study. All samples were centrifuged at 10,000× *g* for 10 min at 4 °C (Hanil, Seoul, Republic of Korea). Thereafter, viral RNA was extracted from 200 μL of each sample using a commercial nucleic acid extraction kit with a fully automated magnetic bead operating platform (TANBead nucleic acid extraction kit, Taiwan Advanced Nanotech Inc., Taoyuan, Taiwan) and then eluted in 100 μL of elution buffer as per the manufacturer’s instructions. All remaining supernatants and nucleic acid samples were stored at −80 °C.

### 2.2. Detection of SOV and other Pathogens

SOV was tested using a previously described real-time reverse transcription polymerase chain reaction (RT-qPCR) assay targeting CPnV [5]. The primers and probe sequences of qRT-PCR were modified to reflect the target gene sequence of a previously reported SOV sequence (GenBank accession number KX364383). The detection primers and probe sequences were forward 5′- AAGATAAATTCTTCTATGAAAACAGAATGA-3′, reverse 5′- CTGCCTAAGTACTATCCAGCCATACTGC-3′, and probe 5′-6-carboxyfluorescein (FAM)-CCATCATAAGTGAGATTTCTAT-Black Hole Quencher 1-3′. The commercially available VDx SIV RT-PCR (MEDIAN Diagnostics Inc., Chuncheon-si, Republic of Korea), Prime-Q PRRSV Detection (GeNet Bio Inc., Daejun, Republic of Korea), and LiliF^®^ Myco-P PCR (Intron Biotechnology, Seongnam-Si, Republic of Korea) kits were used to detect the swine influenza virus (SIV), porcine reproductive and respiratory syndrome virus 1 and 2 (PRRSV-1 and PRRSV-2), and *mycoplasma hyopneumoniae* (MP), respectively. The previously described multiplex qPCR and RT-qPCR methods were used to detect porcine circovirus 2 and 3 (PCV2 and PCV3) and porcine respirovirus 1 (PRV-1), respectively [6,7].

### 2.3. Genome Sequencing of SOV

Of the SOV-positive samples, three samples with the lowest Ct values (<25) were selected for viral genome sequencing. Complete genome sequencing was carried out using 14 pairs of primers, while G gene sequencing was performed using one pair of primers that were newly designed based on the conserved regions of eight complete genome sequences of orthopneumoviruses (five CPnV, two MPV, and one SOV) originating from different hosts and available in GenBank (Appendix A). Each cDNA fragment was synthesized using a commercial kit (Primescript™ 1st strand cDNA Synthesis Kit; Takara Korea Biomedical Inc., Seoul, Republic of Korea), and PCR was performed with synthesized cDNA and the designed primer sets using a commercial kit (PrimeSTAR^®^ GXL DNA Polymerase; Takara Korea Biomedical Inc., Seoul, Republic of Korea) according to the manufacturer’s instructions. The amplified PCR products were purified using a commercial kit (GeneAll Expin™ Combo GP 200 miniprep kit, GeneAll, Seoul, Republic of Korea), and analysis of the sequences of each product was carried out by a commercial company (BIONICS, Daejeon, Republic of Korea) using Sanger’s method in duplicate. As a result, two complete genome sequences (KSOV-2201 and KSOV-2202) and one G gene sequence (KSOV-2203) were obtained and deposited in GenBank as accession numbers OR701947, OR701948, and OR701949, respectively.

### 2.4. Multiple Alignment and Phylogenetic Analysis

For comparative analysis of the sequences, eight complete genome and 24 G gene sequences of SOVs, CPnVs, Feline pneumoviruses (FPnVs), and MPVs were obtained from the GenBank database (accessed on 31 December 2022). The reference sequences of BRSV (Genbank accession: AF295543), HRSV type A (Genbank accession: MW582528), and HRSV type B (Genbank accession: MW582529) were used for the outgroup of complete genome analysis. Multiple sequence alignments were generated using MAFFT, available in Geneious Prime (https://www.geneious.com, accessed on 1 June 2023), and phylogenetic trees were constructed based on the general time-reversible nucleotide substitution with a gamma distribution (GTR GAMMA) model using the RAxML method on Geneious Prime [8,9]. These trees were subjected to bootstrap analysis with 1000 replicates to determine the percentage of reliability for each internal node of the tree.

## 3. Results and Discussion

Of the 389 oral fluid samples collected from 40 pig farms, 14 samples from 4 pig farms located in four provinces of the Republic of Korea were confirmed to be SOV-positive, suggesting that SOV had already spread among some Korean pig farms (Figure 1A). Based on the RT-qPCR results, the sample-level and farm-level detection rates were 4.4% (14/389) and 10% (4/40), respectively. The sample-level detection rate was similar to the previously reported pooled-sample-level detection rate in the United States (5.0%) [2]. However, the detection rate at the farm level was lower than in Spain (29.1%, 16/55). In detail, the SOV detection rate was reported to be 44.4% (12/27) in SIV-positive farms and 14.8% (4/28) in SIV-negative farms [4]. The farms in our study were all SIV-negative, and the SOV detection rate at the farm level was similar to that of the previous study of SIV-negative farms in Spain. This suggests that SOV infection may increase when co-infected with other pathogens, especially SIV. Therefore, it is important to investigate multiple pathogens simultaneously when monitoring swine respiratory pathogens, including SOV.

Further molecular screening for other swine respiratory pathogens showed that PRRSV-1, PRRSV-2, PCV2, PRV-1, and/or MP were detected from most SOV-positive samples (Table 1), suggesting that co-infections of SOV and other respiratory pathogens are common in pig herds in the Republic of Korea. However, the role of SOV in the pathogenesis of porcine respiratory disease complex (PRDC) is uncertain. Therefore, further studies are necessary to investigate the pathological role of SOV in PRDC and assess its clinical impact on affected pig farms.

To obtain more SOV-positive samples, archived blood or tissue samples collected from the four SOV-positive farms were further tested using RT-qPCR, but no more SOV-positive samples were detected (Table 1). These results indicated that SOV may be replicated in the upper respiratory tract (URT) of infected pigs, similar to PRV-1, which was also replicated in limited amounts in the URT of infected pigs and detected only in oronasal swabs and oral fluid samples [10,11,12]. These results suggested that oral fluid or oronasal swab samples are more suitable than blood and tissue samples for the surveillance of SOV infections in pig farms.

Although SOV infections have been reported in the USA, Spain, and France [2,4,13], only one complete genome sequence has been reported from the USA in 2016 [2]. Therefore, viral gene sequencing was carried out to further characterize the Korean SOVs. The 14 pairs of primers designed in this study suitably amplified the viral gene fragments for complete genome sequencing of SOV-positive samples. In this study, we successfully obtained two complete genome sequences and one G gene sequence from SOV-positive samples with the highest viral loads (Table 1).

The complete genome sequence of the KSOV-2201 strain (GenBank accession number OR701947) was 14,881 bp in length, and this was 5 bp shorter than that of the KSOV-2202 strain, which was 14,886 bp in length (GenBank accession number OR701948). A comparison of the two Korean SOVs with the previously reported USA SOV strain 57 (KX364383; 14,885 bp in length) showed slight differences in lengths. The genome was comprised of 11 open reading frames encoding nonstructural proteins 1 and 2 (NS1 and NS2), nucleocapsid protein (N), phosphoprotein (P), membrane protein (M), small hydrophobic protein (SH), attachment glycoprotein (G), fusion protein (F), M2-1, M2-2, and large protein (L), similar to the USA SOV strain [2]. Based on the complete genomic sequences, KSOV-2201 and KSOV-2202 shared 95.7% nt sequence identity. The 5 bp difference in complete genome sequences between two Korean SOV strains was determined to be caused by insertions or deletions in four intergenic regions located between M–SH, SH–G, F–M2-1, and M2-2–L (Appendix A). However, no insertions or deletions were identified in the coding regions. Interestingly, the nucleotide sequences of the 3′-trailer region and the position of the start codon of NS1 in the two Korean SOVs differed from those of USA SOV strain 57 but were similar to other MPVs and MPV-like orthopneumoviruses (MLOV) with different host origins (Appendix A).

Phylogenetic analysis based on the complete genome showed that the KSOV-2201 and KSOV-2202 strains were closely related to MPVs and MLOVs and distantly related to the BRSV and HRSV strains (Figure 1B). This result shows that the orthopneumoviruses currently found in pigs and dogs, including SOV in the Republic of Korea, are phylogenetically more clearly related to MPV than to BRSV or HRSV. Details of the nucleotide identity of the complete genome between Korean SOVs (KSOV-2201 and KSOV-2202) and other eight MLOV strains are shown in Table 2. The complete genome sequences of the KSOV-2201 and KSOV-2202 strains shared 90.3–98.2% nt sequence identity with other MLOV strains and 98.2 and 95.4% homologies with the USA SOV strain 57, respectively. Although the complete genome sequences of MLOVs were limited to ten strains from ten countries, the KSOV-2201 and KSOV-2202 strains were seen to be more closely related to the USA SOV and Chinese CPnV strains than the USA MPV and Italian and Thai CPnV strains. These results suggest that similar to PRV-1, the Korean SOV infections may have been a result of transmission from the US, potentially through the importation of breeding pigs from the country or cross-species transmission from dogs infected with CPnV [10]. The potential risks for the cross-species transmission of the orthopneumovirus between different host species have been raised by previous studies [2,5]. Therefore, further studies are needed to discover the viral dynamics in the Republic of Korea, including viral surveillance in the Korean dog population.

The G protein of orthopneumovirus is a highly variable structural protein that is involved in viral attachment with the host cell surface molecules [14]. It has been used in numerous epidemiological and evolutionary studies in HRSV research [15]. In the current study, the G gene sequences of three Korean SOVs, including the KSOV-2203 strain, were compared with those of 2 MPVs and 22 MLOVs available on GenBank (Figure 2). Previous evidence suggests that the G protein of CPnV and SOV are 18 aa longer than its counterparts in MPVs [2,3,16]. The Korean SOV strains were also seen to exhibit G genes that were 415 aa in length, similar to the USA SOV strain 57 [2]. Phylogenetic analysis based on the G gene sequence showed that the MLOVs were classified into two distinct genogroups, G1 and G2. The pairwise identity of the inter-genogroup was found to be 85.8–88.2%, while the pairwise identity of the intra-genogroup was 90.5–100% and 93.4–100% for G1 and G2, respectively. All four SOV strains (i.e., one from the USA and three from the Republic of Korea) belonged to the G2 group, along with two previously reported Chinese CPnV strains. The G genes of the KSOV-2202 and KSOV-2203 strains exhibited the highest homology (93.9% and 100%, respectively) with the USA 57 strain. In contrast, the G gene of the KSOV-2201 strain exhibited the highest homology (98.6%) with the Chinese CPnV strain SMU-2020-CB19 (Figure 2).

Interestingly, 25 G gene sequences of MLOVs were grouped into two genogroups with 14.2% genetic diversity, and a relationship was observed between the genogroups and the host origin of the viruses. Nineteen CPnV strains were distributed into two genogroups (17 in G1 and two in G2), while two FPnV and four SOV strains were exclusively grouped into G1 and G2, respectively (Figure 2). Although it was difficult to provide irrefutable evidence due to the limited number of viral sequences analyzed in this study, these results suggest ongoing cross-species transmission of MLOVs between dogs, cats, and pigs. Therefore, the SOV may have originated from another host-virus, such as CPnV, which exhibits the highest genetic diversity. No studies have reported pneumovirus infections in dogs and cats in the Republic of Korea to date, highlighting the importance of investigating this across various animal species, including dogs, cats, and pigs.

## 4. Conclusions

This is the first study to report the detection and genetic characterization of SOVs in the Republic of Korea. The SOV-positive pig farms were distributed across four provinces and exhibited co-infections with various respiratory pathogens. We successfully sequenced two complete genomes and one G gene from SOV-positive samples and also conducted phylogenetic and genetic analyses. The Korean SOV strains were related to previously reported SOVs and CPnVs and belonged to genogroup 2 of MLOVs. Complete genome analysis of Korean SOV strains will contribute to expanding our understanding of the genetic characteristics of orthopneumoviruses. Further research is necessary to elucidate the pathogenesis of SOV in PRDC, evaluate the evolutionary pathway of the virus, and confirm cross-species transmission between different host species.

## Figures and Tables

**Figure 1 viruses-15-02371-f001:**
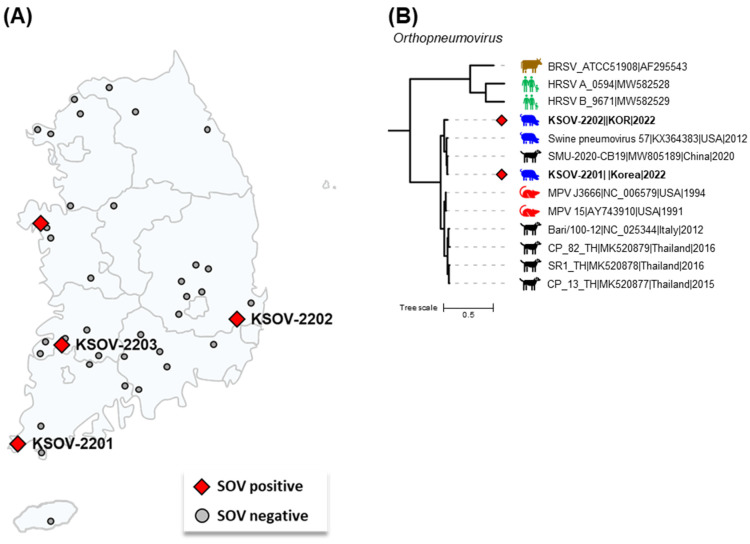
The geographic distribution and complete genome-based phylogenetic tree of Korean swine orthopneumovirus (SOV) strains. (**A**) The locations of the SOV-positive and SOV-negative farms have been shown as red diamonds and gray circles, respectively. The names of sequenced SOV strains have been shown next to the respective farms’ locations. (**B**) The phylogenetic tree was constructed with the complete genomes of MPVs, MPV-like orthopneumoviruses, and the reference strains of BRSV, HRSV type A, and HRSV type B. The two Korean SOV strains are shown as red diamonds. Scale bars indicate nucleotide substitutions per site.

**Figure 2 viruses-15-02371-f002:**
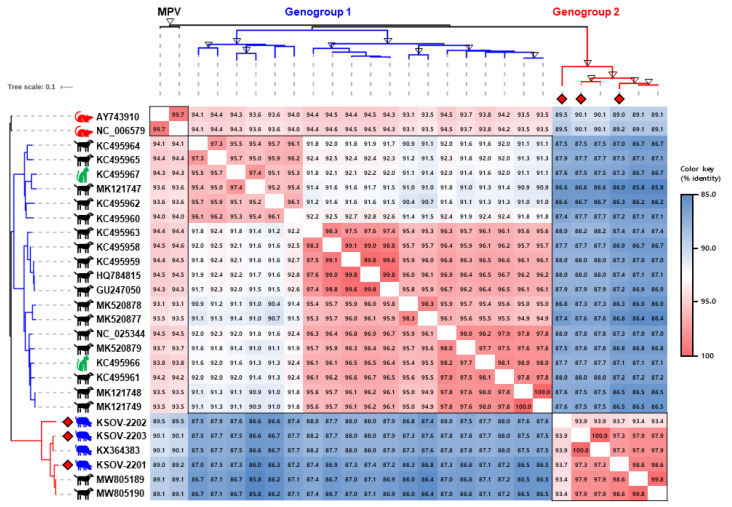
The phylogenetic tree and pairwise identity matrix of the murine pneumovirus (MPV) and MPV-like orthopneumovirus (MLOV) G genes. The phylogenetic tree was constructed using the G gene sequences of three SOV strains identified in this study (red diamonds) and 24 global MLOVs available on GenBank. The branch colors of the tree indicate genogroup 1 (blue) and genogroup 2 (red). Bootstrap values greater than 90 are marked as empty triangles in the corresponding branches. Scale bars indicate nucleotide substitutions per site. The pairwise nucleotide identities of 27 G gene sequences have been shown in various colors per the color key provided on the right side of the figure.

**Table 1 viruses-15-02371-t001:** Characteristics of the samples collected from SOV-positive farms and detection of SOV and respiratory pathogens.

Province	Farm	Age	Sample Type	SOV Detection (Ct Value)	Strain	Detection of Other Respiratory Pathogens
Chungcheongnam-do	CN1	5 weeks	Oral fluid	27.32	-	PRRSV-1
		8 weeks	Oral fluid	37.73	-	PRRSV-1, PRRSV-2
		9 weeks	Oral fluid	35.93	-	-
		4 weeks	Blood	NA	-	-
		8 weeks	Blood	NA	-	PRRSV-1, PRRSV-2
		6 weeks	Lung tissue	NA	-	PRRSV-1, PRRSV-2, PCV2
		6 weeks	Lymphoid tissue	NA	-	PRRSV-1, PRRSV-2, PCV2
Jeollabuk-do	JB2	4 weeks	Oral fluid	25.13	KSOV-2203	PRV-1
		5 weeks	Oral fluid	28.8	-	PRV-1
		6 weeks	Oral fluid	35.78	-	PRV-1
		7 weeks	Oral fluid	38.74	-	PRV-1
		9 weeks	Oral fluid	NA	-	PRV-1
		4 weeks	Lung tissue	NA	-	-
		6 weeks	Lung tissue	NA	-	-
Jeollanam-do	JN5	4 weeks	Oral fluid	NA	-	PRRSV-1, PRV-1
		6 weeks	Oral fluid	24.92	KSOV-2201	PRRSV-2, PRV-1
		7 weeks	Oral fluid	30.37	-	PRRSV-1, PRRSV-2, PRV-1
		8 weeks	Oral fluid	32.18	-	PRRSV-1, PRRSV-2, PRV-1
		9 weeks	Oral fluid	36.57	-	PRRSV-1, PRV-1
		10 weeks	Oral fluid	38.23	-	PRRSV-1, PRV-1
		reserve sows	Oral fluid	NA	-	-
		farrowing sows	Oral fluid	NA	-	PRRSV-1, PRRSV-2
		1 week	Blood	NA	-	-
		2 weeks	Blood	NA	-	-
		3 weeks	Blood	NA	-	PRRSV-2
		4 weeks	Blood	NA	-	-
		5 weeks	Blood	NA	-	-
		6 weeks	Blood	NA	-	PRRSV-2
		7 weeks	Blood	NA	-	PRRSV-2
		8 weeks	Blood	NA	-	PRRSV-2
Gyeongsangbuk-do	GB8	3 weeks	Oral fluid	23.55	KSOV-2202	PCV2, PRV-1, MP
		7 weeks	Oral fluid	29.26	-	PCV2, PRV-1, MP
		9 weeks	Oral fluid	NA	-	PCV2, PRV-1, MP
		13 weeks	Oral fluid	NA	-	PCV2, PRV-1, MP

Abbreviations: SOV—swine orthopneumovirus; PRRSV—porcine reproductive and respiratory syndrome virus; PCV2—porcine circovirus 2; PRV-1—porcine respirovirus 1; MP—*mycoplasma hypneumoniae*; NA—not amplified.

**Table 2 viruses-15-02371-t002:** Comparisons of the complete genome sequences of Korean SOV strains KSOV-2201 and KSOV-2202 with two MPV strains and six MPV-like orthopneumovirus strains.

Host/		Complete	NS1	NS2	N	P	M	SH	G	F	M2-1	M2-2	L
Strain		% Identity to KSOV-2201/KSOV-2202											
Mice/ J3666	nt	92.5/92.3	93.3/92.4	89.0/89.6	94.2/94.1	93.2/93.7	95.2/94.2	93.2/90.3	89.3/89.5	94.0/93.6	94.0/94.4	93.6/93.6	93.8/93.6
	aa		92.1/91.2	93.6/91.7	98.0/98.2	93.6/94.9	99.2/98.1	73.0/71.3	85.9/85.9	96.1/95.5	96.6/96.6	94.9/96.0	97.0/96.9
Mice/ mice_15	nt	92.3/92.2	93.3/92.4	89.0/89.6	94.3/94.2	92.7/93.1	94.8/93.7	90.3/87.5	89.0/89.4	93.9/93.5	94.0/94.4	93.6/93.6	93.8/93.6
	aa		92.1/91.2	93.6/91.7	98.2/98.5	92.6/93.6	98.4/97.3	82.8/80.6	86.3/85.7	96.1/95.5	96.6/96.6	94.9/96.0	97.0/96.8
Pig/Swine_pneumovirus_57	nt	98.2/95.4	91.0/89.5	98.1/94.9	98.8/96.4	98.2/95.7	99.5/96.9	99.6/96.1	97.3/93.9	99.2/97.1	99.4/97.4	99.0/96.3	99.3/96.6
	aa		87.4/84.0	98.7/93.6	99.2/98.7	98.0/97.3	100/98.8	100/95.7	94.9/92.3	99.3/98.1	99.4/98.3	100/96.0	99.4/97.9
Dog/ SMU-2020-CB19	nt	97.4/93.9	98.8/95.6	99.2/94.7	99.5/95.9	99.0/95.0	99.6/97.0	98.9/95.3	98.6/93.4	99.8/97.1	98.5/96.4	99.3/96.0	99.3/96.3
	aa		98.2/91.2	99.4/93.0	100/98.5	98.6/97.0	99.2/98.1	100/95.7	98.1/91.1	100/98.5	98.9/97.2	100/96.0	99.8/98.2
Dog/ Bari/100-12	nt	90.6/90.7	88.6/88.3	85.8/86.0	93.0/92.6	91.8/92.8	93.7/93.1	90.7/89.2	87.4/88.0	92.4/92.2	93.2/93.6	92.9/92.3	92.5/92.7
	aa		88.6/87.7	89.8/87.9	97.2/97.5	92.9/94.9	97.3/96.5	90.3/88.2	83.1/83.4	95.4/95.0	97.7/96.6	93.9/93.9	96.6/96.6
Dog/ CP_13_TH	nt	90.7/90.8	88.6/88.3	87.3/87.5	92.9/92.5	92.0/92.6	93.7/93.0	91.8/91.0	86.9/87.4	92.4/92.1	93.2/93.2	93.3/91.9	92.4/92.6
	aa		87.7/86.8	90.4/88.5	98.2/98.5	92.9/94.6	97.7/96.9	91.4/89.2	82.2/82.9	95.2/94.8	97.7/96.6	93.9/93.9	96.8/96.7
Dog/ CP_82_TH	nt	90.3/90.4	87.4/87.1	85.8/86.4	92.5/92.7	91.0/91.7	92.9/92.2	90.7/89.2	86.9/87.5	91.7/91.5	93.2/93.6	92.9/92.9	92.4/92.5
	aa		85.1/84.2	87.9/86.0	97.7/98.0	91.6/93.6	97.3/96.5	91.4/89.2	81.9/83.1	94.6/94.2	97.7/96.6	93.9/93.9	96.5/96.4
Dog/ SR1_TH	nt	90.5/90.5	87.7/87.4	86.6/86.8	92.9/92.6	91.9/92.5	93.7/93.0	91.8/90.3	86.3/86.8	92.3/91.8	93.0/93.0	93.3/91.9	92.3/92.4
	aa		87.7/86.8	89.2/87.3	98.2/98.5	92.2/93.9	97.7/96.9	91.4/89.2	80.7/81.7	94.8//94.4	97.2/96.0	93.9/93.9	96.3/96.2

Abbreviations: SOV—swine orthopneumovirus; MPV—murine pneumovirus; NS1—nonstructural protein 1; NS2—nonstructural protein 2; N—nucleocapsid protein; P—phosphoprotein; M—membrane protein; SH—small hydrophobic protein; G—attachment glycoprotein; F—fusion protein; L—large protein; nt—nucleotide; aa—amino acid.

## Data Availability

The authors declare that all data supporting the findings of this study are available within the paper. The KSOV-2201, KSOV-2202, and KSOV-2203 genome sequences have been deposited in GenBank under the accession numbers OR701947 to OR701949.

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
