# Peer review of "First Detection and Genetic Characterization of Swine Orthopneumovirus from Domestic Pig Farms in the Republic of Korea"

_viruses, 2023, doi:10.3390/v15122371_

Round 1

Reviewer 1 Report

Comments and Suggestions for Authors

This manuscript reports the first detection of Swine Orthopneumovirus (SOV) in pig from South Korea. In addition to detect SOV RNA in swine oral fluids in pigs from several geographic regions of Korea, the authors expanded their research to archival samples to verify whether tissues from pigs of the SOV positive farms were also SOV positive. Along with all this investigation, the authors were able to obtain the whole genome sequence of two SOV, contributing to the better understanding of the virus genetics and epidemiology. The report is well written and clear, but there is some aspects that can be improved, as listed below.

Page 1, line 26: Clarify what is the prevalence based on (antibodies, genetic material) and more epidemiological information, such as the number of pigs sampled, if all of them were from commercial farms, for instance.

Page I, line 33: What does “FPnVs” stand for?

Page 4, line 54: My understanding that this is the first report of SOV in Asia, which does not mean that other countries are/were not SOV positive. Thus, replace “also the first country in Asia do be affected”, for “also the first country in Asia to report the detection of SOV.

Page 4, 2.1 and S1: Include more information in the table, such as number of pigs sampled per farm, type of farm (farrow-to-finish, wean-to-finish, nursery, etc). Were the farms commercial farms? Or backyard?

Page 4, line 71: The citation for the RT-qPCR stated the article used as a reference is related to the detection of the Canine Orthopneumovirus (Reference #5). Is that correct? If it is, please, clarify how the primers/target RNA region for Canine and Swine Orthopneumovirus are similar.

Page 4, line 79: Did you mean “< 25”?                

Page 3, first paragraph: Were there any similarities among the positive farms or among the positive pigs that could explain the low prevalence? Include this explanation in the text.

Page 3, lines 125 to 126 “such frequent co-infections may lead to the 125 outbreak and exacerbation of PRDV in the field”: This sentence is way too speculative, since the detection of a virus in a sample is just the first step to understand the real involvement of the pathogen with clinical signs. I suggest you remove it.

Page 7, line 225: Replace “indicate” for “suggest”.

Author Response

Dear Reviewer

We would like to express our gratitude to the editor and reviewers for taking time to review our manuscript. Please find enclosed the revised version of our manuscript. We have addressed the reviewers’ comments and have revised the manuscript accordingly. Below are our point-by-point responses to the reviewers’ comments: We hope that you find our revised manuscript suitable for publication in Viruses.

Comments from Reviewer 1

Page 1, line 26: Clarify what is the prevalence based on (antibodies, genetic material) and more epidemiological information, such as the number of pigs sampled, if all of them were from commercial farms, for instance.

Response: As suggested by the reviewer, the relevant sentence was revised as follows (Lines 26-27):

“The detection rate of SOV in oral fluid samples using qRT-PCR was 4.4% (14/389), indicating the presence of the virus in pigs at commercial farms in Korea.”

Page I, line 33: What does “FPnVs” stand for?

Response: Thank you for pointing this out. We revised the abbreviation “FPnVs” to the original name of the virus “feline pneumoviruses” (Line 34).

Page 4, line 54: My understanding that this is the first report of SOV in Asia, which does not mean that other countries are/were not SOV positive. Thus, replace “also the first country in Asia do be affected”, for “also the first country in Asia to report the detection of SOV.

Response: As suggested by the reviewer, the relevant sentence was revised as follows (Lines 55-56):

“The current study presents the first case of SOV infection in domestic pig farms in South Korea which is, incidentally, also the first country in Asia to report the detection of SOV.”

Page 4, 2.1 and S1: Include more information in the table, such as number of pigs sampled per farm, type of farm (farrow-to-finish, wean-to-finish, nursery, etc). Were the farms commercial farms? Or backyard?

Response: As suggested by the reviewer, we have included number of pig samples in Table S1 and detailed the information of farms as follows (line 61-63):

“In 2022, 389 oral fluid samples were collected from 40 commercial farrow-to-finish pig farms suffering from respiratory distress in nine provinces in South Korea, as part of the porcine reproductive and respiratory syndrome virus (PRRSV) surveillance project. Table S1 summarizes the characteristics of the pig farms examined in this study.”

Page 4, line 71: The citation for the RT-qPCR stated the article used as a reference is related to the detection of the Canine Orthopneumovirus (Reference #5). Is that correct? If it is, please, clarify how the primers/target RNA region for Canine and Swine Orthopneumovirus are similar.

Response: Thanks for reviewer’s kind and scientific comments. We have addressed errors in the reference paper within the method section and provided additional details to address insufficient explanations regarding primer information (line 72-78).

“SOV was tested using a previously described real-time reverse transcription polymerase chain reaction (RT-qPCR) assay targeting CPnV [5]. The primers and probe sequences of qRT-PCR were modified to reflect the target gene sequence of previously reported SOV sequence (GenBank accession number KX364383). The detection primers and probe sequences were: forward 5′- AAGATAAATTCTTCTATGAAAACAGAATGA-3′, reverse 5′- CTGCCTAAGTACTATCCAGCCATACTGC-3′, probe 5′-6-carboxyfluorescein (FAM)-CCATCATAAGTGAGATTTCTAT-Black Hole Quencher 1-3′.”

Page 4, line 79: Did you mean “< 25”?               

Response: Reviewer is correct. We revised the “25<” to “<25” (Line 88).

Page 3, first paragraph: Were there any similarities among the positive farms or among the positive pigs that could explain the low prevalence? Include this explanation in the text.

Response: Thanks for reviewer’s scientific comment. Given that we used samples collected to investigate PRRSV and that the age at which samples were collected was not uniform across farms, we have determined that the term "detection rate" would be more appropriate for this study rather than using the term "prevalence". Additionally, we referenced a report on the prevalence of SOV in Spain to describe our low prevalence rate. Also, we emphasized the necessity for a comprehensive investigation of porcine respiratory pathogens in Korea. We revised the relevant sentence (line 119-129) as follows:

“Based on the RT-qPCR results, the sample-level and farm-level detection rates were 4.4% (14/389) and 10% (4/40), respectively. The sample-level detection rate was similar to the previously reported pooled-sample-level detection rate in the United States (5.0%) [2]. However, the detection rate at the farm-level was lower than in Spain (29.1%, 16/55). In detail, the SOV detection rate was reported to be 44.4% (12/27) in SIV-positive farms and 14.8% (4/28) in SIV-negative farms [4]. The farms in our study were all SIV-negative, and the SOV detection rate at the farm-level was similar to that of the previous study of SIV-negative farms in Spain. This suggests that SOV infection may increase when co-infected with other pathogens, especially SIV. Therefore, it is important to investigate multiple pathogens simultaneously when monitoring swine respiratory pathogens, including SOV.”

Page 3, lines 125 to 126 “such frequent co-infections may lead to the 125 outbreak and exacerbation of PRDV in the field”: This sentence is way too speculative, since the detection of a virus in a sample is just the first step to understand the real involvement of the pathogen with clinical signs. I suggest you remove it.

Response: As recommended by the reviewer, we removed the sentence.

Page 7, line 225: Replace “indicate” for “suggest”.

Response: As suggested by the reviewer, we revised “indicate” to “suggest” (Line 242).

We thank the editor and reviewers for their thoughtful comments on our manuscript.

Sincerely,

Seung-Chun Park

Laboratory of Veterinary Pharmacokinetics and Pharmacodynamics

Institute for Animal Biomedical Science

College of Veterinary Medicine

Kyungpook National University

Daegu 41566, Republic of Korea

November 23, 2023

Reviewer 2 Report

Comments and Suggestions for Authors

SOV has not yet been isolated, and its prevalence and pathogenicity are unclear. The current study presents the first case of SOV infection in domestic pig farms in South Korea. In this study genetic characterization of the Korean SOV strain was carried. Due to the fact that data on the distribution of orthopneumoviruses in different countries are limited, this study is interesting and relevant. The article is well written, and the methodology used is adequate. The article is interesting and worthy of publication, however there are some questions:

- You noted that no studies have reported pneumovirus infections in dogs and cats in Korea to date. Are mass diagnostic tests for the presence of orthopneumoviruses in samples collected from cats and dogs in South Korea?

- Why weren’t serological studies (by ELISA) carried out to better understand the spread of infection?

- Have any clinical signs of respiratory infections been noted in positive animals? If possible, add some detail about the symptoms of infection on farms, especially for animals with SOV without other pathogens. Perhaps this information will be useful for studying the role of this virus in co-infections with various respiratory pathogens.

- Your results on co-infections SOVs with various respiratory pathogens confirm results obtained in other countries. However, these studies do not include studies of co-infection with bacterial pathogens. In my opinion, these studies would be interesting in the future.

Author Response

Dear Reviewer

We would like to express our gratitude to the editor and reviewers for taking time to review our manuscript. Please find enclosed the revised version of our manuscript. We have addressed the reviewers’ comments and have revised the manuscript accordingly. Below are our point-by-point responses to the reviewers’ comments: We hope that you find our revised manuscript suitable for publication in Viruses.

Comments from Reviewer 2

- You noted that no studies have reported pneumovirus infections in dogs and cats in Korea to date. Are mass diagnostic tests for the presence of orthopneumoviruses in samples collected from cats and dogs in South Korea?

Response:

Our research team is developing a diagnostic method for companion animal pathogens and continuously collecting samples from dogs and cats. Respiratory pathogens, including orthopneumovirus, will be investigated.

- Why weren’t serological studies (by ELISA) carried out to better understand the spread of infection?

Response: To the best of our knowledge, the only commercially available ELISA kits are for RSV, BRSV, and MPV. Therefore, our research team is developing an In-house ELISA or IFA method for the accurate serological study of SOV. Before this, we submitted our first SOV detection paper for rapid reporting.

- Have any clinical signs of respiratory infections been noted in positive animals? If possible, add some detail about the symptoms of infection on farms, especially for animals with SOV without other pathogens. Perhaps this information will be useful for studying the role of this virus in co-infections with various respiratory pathogens.

Response: Thanks for reviewer’s scientific comment.

The samples were submitted to our laboratory from farms experiencing respiratory symptoms, but unfortunately, we were unable to gather detailed information on the clinical symptoms for each individual farm. Additionally, because there were no farms with SOV-only infection, clinical symptoms caused by SOV could not be identified. Therefore, our research team will attempt to isolate SOV using cell culture, and will use the isolated virus to conduct research on clinical symptoms caused by SOV infection alone or co-infection with other pathogens.

- Your results on co-infections SOVs with various respiratory pathogens confirm results obtained in other countries. However, these studies do not include studies of co-infection with bacterial pathogens. In my opinion, these studies would be interesting in the future.

Response: We agree with the reviewer's opinion. In accordance with the reviewer's opinion, we will also incorporate monitoring for bacterial infections in the upcoming investigation of pig respiratory pathogens. Additionally, we conducted tests targeting samples from SOV-positive farms for Mycoplasma hyopneumoniae (MP), a key bacterium causing respiratory diseases in pigs. The result showed that one SOV-positive farm (GB8) was co-infected with MP. We presented these results in Material and Method (Line 78-83) and Result and Discussion sections (Line 130-133).

“The commercially available VDx SIV RT-PCR (MEDIAN Diagnostics Inc., South Korea), Prime-Q PRRSV Detection (GeNet Bio Inc., Daejun, Korea), and LiliF® Myco-P PCR (Intron Biotechnology, Korea) kits were used to detect the swine influenza virus (SIV), porcine reproductive and respiratory syndrome virus 1 and 2 (PRRSV-1 and PRRSV-2), and mycoplasma hyopneumoniae (MP) respectively.”

“Further molecular screening for other swine respiratory pathogens showed that PRRSV-1, PRRSV-2, PCV2, and/or PRV-1, and/or MP were detected from most SOV-positive samples (Table 1), suggesting that co-infections of SOV and other respiratory pathogens are common in pig herds in Korea.”

We thank the editor and reviewers for their thoughtful comments on our manuscript.

Sincerely,

Seung-Chun Park

Laboratory of Veterinary Pharmacokinetics and Pharmacodynamics

Institute for Animal Biomedical Science

College of Veterinary Medicine

Kyungpook National University

Daegu 41566, Republic of Korea

November 23, 2023
